# Exosomes from Human Periodontal Ligament Stem Cells Promote Differentiation of Osteoblast-like Cells and Bone Healing in Rat Calvarial Bone

**DOI:** 10.3390/biom14111455

**Published:** 2024-11-17

**Authors:** Mhd Safwan Albougha, Hideki Sugii, Orie Adachi, Bara Mardini, Serina Soeno, Sayuri Hamano, Daigaku Hasegawa, Shinichiro Yoshida, Tomohiro Itoyama, Junko Obata, Hidefumi Maeda

**Affiliations:** 1Department of Endodontology and Operative Dentistry, Faculty of Dental Science, Kyushu University, 3-1-1 Maidashi, Higashi-ku, Fukuoka 812-8582, Japan; albougha.mhd.246@s.kyushu-u.ac.jp (M.S.A.);; 2Department of Endodontology, Kyushu University Hospital, 3-1-1 Maidashi, Higashi-ku, Fukuoka 812-8582, Japan

**Keywords:** periodontal ligament stem cells, exosomes, osteoblast-like cells, bone healing

## Abstract

Deep caries and severe periodontitis cause bone resorption in periodontal tissue, and severe bone resorption leads to tooth loss. Periodontal ligament stem cells (PDLSCs) are important for the healing of defective periodontal tissue. It is increasingly understood that healing of periodontal tissue is mediated through the secretion of trophic factors, particularly exosomes. This study investigated the effects of exosomes from human PDLSCs (HPDLSCs-Exo) on human osteoblast-like cells in vitro and on the healing of rat calvarial bone defects in vivo. HPDLSCs-Exo were isolated and characterized by their particle shape, size (133 ± 6.4 nm), and expression of surface markers (CD9, CD63, and CD81). In vitro results showed that HPDLSCs-Exo promoted the migration, mineralization, and expression of bone-related genes such as alkaline phosphatase (*ALP*), bone morphogenetic protein 2 (*BMP2*), osteocalcin (*OCN*), and osteopontin (*OPN*) in human osteoblast-like cells. Furthermore, in vivo results showed that more newly formed bone was observed in the HPDLSCs-Exo-treated group than in the non-treated group at the defect sites in rats. These results indicated that HPDLSCs-Exo could promote osteogenesis in vitro and in vivo, and this suggests that HPDLSCs-Exo may be an attractive treatment tool for bone healing in defective periodontal tissue.

## 1. Introduction

Deep caries and severe periodontitis represent a clinical challenge, especially with aging populations around the world [1,2]. These diseases cause irreversible destruction of the periodontal tissue, which is a complex structure supporting teeth and is composed of periodontal ligament (PDL), cementum, and alveolar bone. In cases with poor prognoses, they can often lead to tooth loss [3,4]. The tooth loss is primarily due to the resorption of supporting bone. Regeneration of resorped bone is essential for the regeneration of defective periodontal tissue. Surgical and nonsurgical treatments for periodontitis are effective in managing the condition; however, these treatments are not sufficient to regenerate the lost periodontal tissue—including alveolar bone—in many clinical situations, and new treatments are needed [5].

Mesenchymal stem cells (MSCs) have shown great potential as a treatment for healing periodontal bone defects both in animal [6,7] and human studies [8]. Dental tissues are an important source of stem cells, and dental stem cells (DSCs) are becoming increasingly vital in the field of tissue engineering [9]. Periodontal ligament stem cells (PDLSCs) are a type of DSC that shows good promotion of healing and multipotency compared with other DSCs, such as dental pulp stem cells and dental follicle progenitor cells [10,11]. Some animal studies have reported that transplanting PDLSCs leads to improved bone healing in periodontal defects [12,13,14] and calvarial bone defects [15,16].

Transplanting MSCs at the injury site is a primary strategy and numerous research efforts have concentrated on utilizing cell therapy to improve the healing of defective bone tissue. However, implementing this type of therapy presents challenges and limitations due to the difficulty of maintaining optimal cell differentiation potency and viability during expansion, growth, and delivery to the patient. In addition, transplanted MSCs do not survive for a long time, leading to limited engraftment [17].

Recent studies have revealed that the potential of MSCs in tissue healing is largely related to their paracrine effects through the secretion of various factors such as growth factors, cytokines, chemokines, enzymes, and extracellular vesicles including exosomes [18,19]. Exosomes are small organelles, ranging from approximately 40 to 160 nm in diameter, with a nanoscale lipid bilayer. They contain many components of the cell, such as proteins, DNA, RNA, lipids, and glycoconjugates [20,21].

Exosomes have various functions, including transmitting information, discarding intracellular components, and acting as drug carriers through the transportation of exogenous chemicals and biomolecules. Furthermore, exosomes secreted from MSCs (MSCs-Exo) have biological functions similar to the MSCs themselves, and they are pivotal mediators of biological functions in MSCs [22]. MSCs-Exo also have many potential therapeutic advantages such as small size, low complexity, no neoplastic transformation risk, increased stability, easy production, and long preservation [23]. These advantages explain why the use of MSCs-Exo is emerging as a promising alternative to cell therapy for tissue regeneration [24,25].

Several studies have reported the therapeutic effects of MSCs-Exo during bone healing [22,26]. It has been reported that various types of MSCs-Exo derived from bone marrow cells [27], adipose cells [28], and dental follicle cells [29] exhibited healing effects on bone defects in periodontal tissue. These MSCs-Exo could promote bone healing but the therapeutic effects on bone volume in animal studies remain unsatisfactory.

Recently, one animal study demonstrated that the conditioned medium from PDLSCs exhibited periodontal healing including the formation of alveolar bone in a concentration-dependent manner [30]. Furthermore, it was reported that exosomes derived from PDLSCs (PDLSCs-Exo) were involved in the tissue healing process for nerve repair [31,32], and the inclusion of polyethylenimine-engineered PDLSCs-Exo within a collagen membrane enhanced the healing and vascularization at bone defect sites [33,34]. However, the effects of exosomes secreted from human PDLSCs (HPDLSCs-Exo) on the biological properties of osteoblast-like cells and healing of bone defects have not been studied in detail.

Therefore, this study evaluated the effects of HPDLSCs-Exo on the proliferation, migration, and differentiation of osteoblast-like cells in vitro as well as bone healing in a rat model with surgically created calvarial bone defects in vivo as we hypothesized that HPDLSCs-Exo can be a new therapeutic material for bone healing.

## 2. Materials and Methods

### 2.1. Cell Culture

A clonal cell line of heterogeneous, immortalized human PDL cells (2-23), which was previously established by our group [35], was used for HPDLSCs. Saos2 cells (RIKEN, Saitama, Japan) were used for human osteoblast-like cells. All cells were cultured in alpha-minimum essential medium (α-MEM; Gibco-BRL, Grand Island, NY, USA) containing 10% fetal bovine serum (FBS; Sigma-Aldrich, St. Louis, MO, USA), 50 µg/mL of streptomycin, and 50 U/mL of penicillin (10% FBS/α-MEM). The cells were cultured at 37 °C in a humidified atmosphere with 5% CO_2_. The absence of mycoplasma contamination was confirmed using Takara PCR Mycoplasma Detection Set (Takara Bio Inc., Shiga, Japan). The result was shown in Appendix A.

### 2.2. Isolation of HPDLSCs-Derived Exosomes

HPDLSCs (1 × 10^5^ cells per well) were seeded on 6-well plates using 10% FBS/α-MEM. After 24 h of culture, the cells were washed with PBS twice, and the culture medium was changed to α-MEM containing 10% exosome-depleted FBS (Exo-FBS; System Biosciences, LLC, Palo Alto, CA, USA). After incubation for 48 h, the supernatant was collected and centrifuged at 2000× *g* for 30 min to remove the cells and debris. The supernatant containing the cell-free culture medium was transferred to a new tube and the pellet was discarded. Total exosome isolation reagent (Invitrogen, Waltham, MA, USA) was then added to the collected supernatant and mixed well. The mixture was incubated at 4 °C overnight before centrifugation at 10,000× *g* for 1 h at 4 °C. The supernatant was aspirated and discarded. The pellet was resuspended in sterile PBS and stored at −20 °C until use. These samples were used as the exosomes from HPDLSCs (HPDLSCs-Exo). The protein concentration of HPDLSCs-Exo was quantified using a BCA Protein Assay kit (Takara Bio Inc., Shiga, Japan). For characterization of HPDLSCs-Exo, the ultrastructure of HPDLSCs-Exo was observed using a transmission electron microscope, the particle size was analyzed with a nanoparticle tracking analysis system, and the expression of exosomal surface markers (CD9, CD63, and CD81) was detected by Western blotting analysis. The original images of Western blotting analysis were shown in Appendix A.

### 2.3. Transmission Electron Microscopy (TEM) Analysis

TEM was used to observe the ultrastructure of HPDLSCs-Exo. A 10-microliter exosomal suspension was sent to an outside contractor (FUJIFILM Wako Pure Chemical Industries Ltd., Osaka, Japan) for TEM analysis. The images were captured using an H-7600 transmission electron microscope at 100 kV (HITACHI Ltd., Tokyo, Japan).

### 2.4. Nanoparticle Tracking Analysis (NTA)

NTA was performed using a NanoSight LM10 (FUJIFILM Wako Pure Chemical Industries Ltd.) to analyze particle size and concentration by visualizing the Brownian motion of nanoparticles in a solvent. The analysis was outsourced to an outside contractor (FUJIFILM Wako Pure Chemical Industries Ltd.). For the Brownian motion observation, the camera level was set to 13; 60-second videos were taken five times; and the particle size and particle concentration were calculated from the video analysis.

### 2.5. Expression of Exosomal Surface Markers

Expression of exosomal surface markers (CD9, CD63, and CD81) was analyzed by Western blotting analysis. The exosome proteins were loaded onto 10% sodium dodecylsulfate polyacrylamide gels for electrophoresis and then were transferred to Immune-Blot polyvinylidene difluoride membranes (Bio-Rad Laboratories, Hercules, CA, USA). The membranes were blocked with no-fat milk for 1 h and then incubated with one of the following antibodies: rabbit monoclonal anti-CD9 antibody (ab92726; Abcam, Cambridge, UK) at a dilution of 1:1000; mouse monoclonal anti-CD63 antibody (012-27063; FUJIFILM Wako Pure Chemical Industries Ltd.) at a dilution of 1:1000; or mouse monoclonal anti-CD81 (MA5-13548; Invitrogen) antibody at a dilution of 1:1000. A biotinylated anti-mouse IgG (Nichirei Biosciences Inc., Tokyo, Japan) or anti-rabbit IgG (Nichirei Biosciences) was added to the membranes and images of the membranes were captured by the ECL Select Western blotting detection system (GE Healthcare, Chicago, IL, USA). The standard protein, BLUeye Prestained Ladder (Gene Direx Inc., Las Vegas, NV, USA), was used in this study. Exo-FBS and PBS were used as the negative control.

### 2.6. Cell Migration Assay

Based on the previous study [36,37], a scratch-wound healing assay was performed to assess the effect of PDLSCs-Exo on the migration of pre-osteoblasts. Briefly, Saos2 cells (3 × 10^4^ cells per well) were seeded on 12-well plates in 10% FBS/α-MEM and were cultured until 90% confluent. The cells were then treated with 1 μg/mL of Mitomycin C (Nacalai Tesque, Kyoto, Japan) for 1 h to stop proliferation. After that, 200 μL micropipette tips were used to scratch across the diameter of the wells to create wounds with similar widths (400 μm wide). The cells were then washed gently with PBS three times to remove detached cells and cultured in culture medium (10% Exo-FBS/α-MEM) with or without 2 or 5 µg/mL of HPDLSCs-Exo. For each well, the same field of view was selected for recording by microscopy and images were taken at 0 h, 24 h, and 48 h after wounding. The numbers of migrated cells in the wounded areas (inside of the yellow dotted lines) were manually counted and averaged from four random fields per well (*n* = 4).

### 2.7. Proliferation Assay

Saos2 cells (3 × 10^3^ cells per well) were seeded on 48-well plates in 250 μL of culture medium (10% Exo-FBS/α-MEM) with or without 1 or 2 µg/mL of HPDLSCs-Exo and were cultured for 3 days. The proliferation of Saos2 cells was measured on days 0, 1, 2, and 3 of culture using the WST-1 Cell Proliferation Assay kit (Millipore Corp., Billerica, MA, USA). At the indicated time points, 25 μL of the WST-1 reagent was mixed with the culture medium and the mixtures were incubated at 37 °C for 30 min. Supernatants of the mixtures (100 μL) were collected from each well and the absorbance at 450 nm was measured using an iMark microplate reader (Bio-Rad Laboratories, Berkeley, CA, USA).

### 2.8. Mineralization Assay

Based on previous studies [38,39], we used calcium chloride (CaCl_2_) for the induction of mineralization. Saos2 cells (1 × 10^4^ cells per well) were cultured on 24-well plates in three different medium types: 10% Exo-FBS/α-MEM (control), control containing 1 mM CaCl_2_ (Ca), or Ca containing 1 or 2 µg/mL of HPDLSCs-Exo (PDLSCs-Exo). Half of the medium in each well was exchanged every 3 days during the mineralization process. After 2 weeks of culture, mineralization of Saos2 cells was analyzed using von Kossa staining or Alizarin red S staining. Briefly, for von Kossa staining, the cells were fixed with 4% paraformaldehyde (PFA, Merck, Darmstadt, Germany) for 40 min and were washed with distilled water. The positive reaction was then visualized with a 2% silver nitrate solution (Nacalai Tesque) as in our previous report [40]. For Alizarin red S staining, the cells were washed with PBS and were fixed with a 10% formalin solution (FUJIFILM Wako Pure Chemical Industries Ltd.) for 1 h before being washed with distilled water and stained with Alizarin Red S (Sigma-Aldrich). The Alizarin red S-positive areas and the von Kossa-positive areas were quantified by a Biozero digital microscope (Keyence Corporation, Osaka, Japan). Using cells grown under the same culture conditions, total RNA was isolated at 3, 7, and 14 days of culture, and the gene expression of bone-related markers was examined.

### 2.9. Quantitative RT-PCR Analysis

Total RNA was isolated with TRIzol reagent (Invitrogen, Carlsbad, CA, USA). First-strand cDNA was synthesized from 1 µg of the total RNA using an ExScript RT Reagent kit (Takara Bio Inc., Shiga, Japan) in accordance with the manufacturer’s instructions. Reverse transcription of total RNA was performed with random 6-mers and ExScriptRTase for 15 min at 42 °C, followed by incubation for 2 min at 99 °C and 5 min at 5 °C to stop the reaction. A KAPA SYBR FAST qPCR Kit (Kapa Biosystems Inc., Boston, MA, USA) was used for PCR under the following conditions: 95 °C for 10 s (initial denaturation), then 40 cycles of 95 °C for 5 s and 60 °C for 30 s, followed by a dissociation program of 95 °C for 15 s, 60 °C for 30 s, and 95 °C for 15 s in a Thermal Cycler Dice Real-Time System (Takara Bio Inc., Shiga, Japan). β-actin was used as the internal control. The expression levels of the target genes were calculated using 2^−ΔΔCt^ values. Primer sequences, annealing temperatures, cycle numbers, and product sizes for *alkaline phosphatase* (*ALP*), *bone morphogenetic protein 2* (*BMP2*), *osteocalcin* (*OCN*), *osteopontin* (*OPN*), *runt-related transcription factor 2* (*RUNX2*), and *β-actin* are shown in Table 1. The GenBank database was used for designing primer sequences, and specificity of the primer sequences was ensured by performing a BLAST search of GenBank. Normalization of the target genes was performed against β-actin expression, and the values were shown as the fold increase relative to the control.

### 2.10. Calvarial Bone Defect Model in Rats

Sprague Dawley rats (12 weeks old, male) weighing 300–350 g (*n* = 6) were used in this study (Kyudo, Saga, Japan). The rats had unrestricted access to food and water throughout the experimental period. The surgical procedure was performed according to a previous report [41]. Anesthesia was induced by intraperitoneal injection of 0.15 mg/kg medetomidine (Kyoritsu Seiyaku Co., Ltd., Tokyo, Japan) along with 2.5 mg/kg butorphanol tartrate (Meiji Seika Pharma Co., Ltd., Tokyo, Japan) and 2 mg/kg midazolam (Sandoz Inc., Tokyo, Japan). An incision was made down to the periosteum and the periosteum was peeled away to expose the calvarium. A critical-sized calvarial defect with a 5-millimeter diameter was created on each side of the calvarial bone using a dental trephine bur (Muromachi Kikai Co., Ltd., Tokyo Japan). The size of calvarial bone defects was determined based on a previous assessment of critical size in rat calvarial bone [42]. To mix HPDLSCs-Exo with hydrogel, HPDLSCs-Exo at a concentration of 60 µg/mL was added to liquid VitroGel RGD-PLUS (The Well Bioscience, North Brunswick, NJ, USA) in a 1:1 ratio. The mixture was injected into the defect site using a micropipette. For the control group, VitroGel RGD-PLUS mixed with only PBS (1:1) was injected. For each rat, both defects received HPDLSCs-Exo treatment or the control treatment. After the injection of HPDLSCs-Exo, the skin was closed over the periosteum using a simple interrupted suture with black silk thread. Six weeks after the surgery, the animals were sacrificed by transcardial perfusion with 4% paraformaldehyde (PFA) (FUJIFILM Wako Pure Chemical Industries Ltd.) in PBS while under anesthesia as described above. Craniums were removed and immersed in 4% PFA for 12 h. The tissues were then washed with PBS and subjected to micro-CT scanning before being decalcified in 10% ethylenediaminetetraacetic acid (FUJIFILM Wako Pure Chemical Industries Ltd.) at 4 °C for 1 month. After dehydration, samples were embedded in optimal cutting temperature compound (Sakura Finetek, Tokyo, Japan). All procedures were performed under the approval of the Animal Ethics Committee and followed the regulations of Kyushu University (approval code: A23-055-0).

### 2.11. Micro-CT Analysis

The samples were scanned by micro-CT (Skyscan 1076; Bruker Corp., Ltd., Billerica, MA, USA) and the images were reconstructed using NRecon software (version 1.6.9.8; Skyscan; Bruker Corp., Billerica, MA, USA). The reconstructed images were evaluated based on the grayscale intensity of the samples. The volume of newly formed bone areas was detected and quantified as in a previous report [43].

### 2.12. Histological Analysis

The frozen samples were sectioned at a thickness of 6 µm for hematoxylin–eosin (HE) staining and Masson’s trichrome staining. HE staining and Masson’s trichrome staining were performed according to the standard protocol [44,45]. The sections were observed using a Biozero digital microscope (Keyence Corporation, Osaka, Japan). Furthermore, the areas of newly formed bone at the defect sites in Masson’s trichrome staining (blue) were quantified using a Biozero digital microscope (Keyence Corporation).

### 2.13. Statistical Analysis

All experiments were analyzed in triplicate or quadruplicate. All values are expressed as mean  ±  standard deviation. Statistical analysis was performed using one-way ANOVA followed by the Benjamini–Hochberg procedure. Statistical significance was determined as a probability value of *p* < 0.05.

## 3. Results

### 3.1. Isolation of Exosomes from HPDLSCs

To confirm whether the supernatant of culture medium from HPDLSCs contains exosomes, the isolated extracellular vesicles were characterized by TEM analysis, NTA, and the expression of exosomal surface markers (CD9, CD63, and CD81). TEM analysis showed the circular morphology of the isolated extracellular vesicles with a size of approximately 100 nm (Figure 1A). NTA showed that the mean value of particles from isolated extracellular vesicles was 133 ± 6.4 nm and the mode value (the most commonly occurring value) was 94 ± 21 nm (Figure 1B). Western blotting analysis demonstrated that these isolated extracellular vesicles expressed CD9, CD63, and CD81, which are exosomal surface markers (Figure 1C). These results indicated that extracellular vesicles were secreted by HPDLSCs, and we could successfully collect exosomes from HPDLSCs (HPDLSCs-Exo).

### 3.2. HPDLSCs-Exo Enhanced the Migration of Saos2 Cells

The effects of HPDLSCs-Exo on the migration of Saos2 cells were analyzed by scratch-wound healing assay. After 24 h and 48 h of culturing, more migrated cells were observed in the 2 µg/mL and 5 µg/mL HPDLSCs-Exo group than in the Cont group (Figure 2A). The quantification of migrated cells showed that 2 µg/mL and 5 µg/mL HPDLSCs-Exo significantly promoted the migration of Saos2 cells compared with the Cont after 24 h and 48 h of culture (Figure 2B and Appendix A). 

The proliferative ability of HPDLSCs-Exo-treated Saos2 cells was analyzed using a WST-1 proliferation assay. The proliferation of Saos2 cells was not affected by HPDLSCs-Exo (Appendix A), indicating that HPDLSCs-Exo can effectively enhance the migration of Saos2 cells but have no effect on their proliferation.

### 3.3. HPDLSCs-Exo Promoted Osteoblastic Differentiation of Saos2 Cells

Saos2 cells were treated with different concentrations of HPDLSCs-Exo (0, 1, and 2 µg/mL) for 2 weeks. The mineralization of Saos2 cells treated with HPDLSCs-Exo was evaluated by Alizarin red S staining (Figure 3A) and von Kossa staining (Figure 3B). The quantification of the positively stained area in both tests showed that HPDLSCs-Exo significantly enhanced the mineralization of Saos2 cells in a dose-dependent manner (Figure 3A,B).

The expression of bone-related genes (*ALP*, *BMP2*, *OCN*, *OPN*, and *RUNX2*) was analyzed at three time points during culturing (3 days, 1 week, and 2 weeks) using quantitative RT-PCR analysis. The expression of *ALP* and *OCN* were significantly increased after 3 days and 1 week of culture, respectively, and expression of *OPN* and *RUNX2* were significantly upregulated after 2 weeks of culture in the 2 µg/mL HPDLSCs-Exo group compared with the Ca group (culturing medium containing 1 mM CaCl_2_) (Figure 3C–E). Furthermore, the expression of *BMP2* was significantly higher in the 2 µg/mL HPDLSCs-Exo group than in the Ca group at all time points (Figure 3C–E). These results demonstrated that 2 µg/mL of HPDLSCs-Exo significantly promoted differentiation of Saos2 cells at all stages of osteoblastic differentiation.

### 3.4. HPDSLCs-Exo Induced the Healing of Bone Defects in Rat Calvarial Bone

To investigate the functions of HPDLSCs-Exo in bone healing, a rat calvarial bone defect model was used in this study. The HPDLSCs-Exo group was treated with HPDLSCs-Exo (60 μg/mL) for 6 weeks. Rats treated with PBS were used as the control group. Micro-CT images showed that there was more newly formed bone at the bone defect sites in the HPDLSCs-Exo group than in the control group (Figure 4A,B and Appendix A). Defect areas in the HPDLSCs-Exo group were almost filled by newly formed bone with several high-density spots (Figure 4B) while the defect areas in the control group had many void areas and small peninsulas of bone along the margins of the defect sites (Figure 4A). The areas of newly formed bone were quantified and more bone volume was detected in the HPDLSCs-Exo group than in the control group (Figure 4C).

Next, anatomical structures in newly formed bone areas were assessed by HE staining and Masson’s trichrome staining. Histological analysis by HE staining showed more well-formed, bone-like tissue in the HPDLSCs-Exo group than in the control group (Figure 5A–D). Furthermore, Masson’s trichrome staining showed that there were more bone-like structures in the HPDLSCs-Exo group than in the control group (Figure 5E–I). The control group exhibited more fibrous structures in the void areas, which were also observed in micro-CT analysis of the defect sites (Figure 5E,G). These results suggested that HPDLSCs-Exo could effectively promote bone healing in rat calvarial bone defects.

## 4. Discussion

In the present study, the biological function of HPDLSCs-Exo was assessed using in vitro and in vivo experiments. HPDLSCs-Exo promoted the migration and osteoblastic differentiation of human osteoblast-like cells and bone healing of rat calvarial bone defects. These results support our hypothesis that HPDLSCs-Exo are useful for the healing of bone defects.

Exosomes are extracellular vesicles secreted by cells with a size of 40–160 nm in diameter and can play crucial roles in physiological and pathological processes by mediating intracellular communication and the transport of cargo [46]. Exosomes are a specific type of extracellular vesicle, so the following analyses are needed to determine whether isolated components are exosomes: TEM for the identification of exosome morphology; NTA for the identification of exosome size; and Western blotting for the identification of exosomal surface markers [47]. Our results demonstrated that HPDLSCs-Exo exhibited a circular morphology, a particle size of 133 ± 6.4 nm, and expression of exosomal surface markers (CD9, CD63, and CD81) using TEM, NTA, and Western blotting analyses, respectively. These characteristics were consistent with the typical description of exosomes [20], indicating that HPDLSCs-Exo were successfully isolated from the culture medium of HPDLSCs.

When periodontal tissue is defective, there is destruction of tissue homeostasis with excessive inflammation resulting in bone loss. To facilitate bone healing and repair, it is important to enhance the proliferation and migration of cells at the defect sites [48]. It was reported that MSCs-Exo had positive effects on the proliferation and migration of various cell types such as osteoblasts, osteocytes, MSCs, and endothelial cells [49]. In our present study, HPDLSCs-Exo promoted the migration of osteoblast-like cells but showed no effect on proliferation. PDLSCs possess a high degree of multipotency with the capacity to differentiate into osteoblasts, adipocytes, chondrocytes, fibroblasts, and cementoblasts [50]. In addition, some reports demonstrated that differentiated daughter cells from MSCs regulated the proliferation of MSCs [51], and MSCs-Exo promoted osteoblastic differentiation of osteoblast-like cells rather than their proliferation [52]. These results indicated that HPDLSCs-Exo are better able to induce osteoblastic differentiation of osteoblast-like cells than to promote their proliferation.

The osteogenesis occurs by osteoblastic differentiation of MSCs. During the osteoblastic differentiation of MSCs, they are differentiated into pre-osteoblasts, which are intermediate cells that can express ALP, an early-stage marker of osteogenesis [53,54,55]. After the formation of newly synthesized bone-like cells, OCN and OPN are expressed in the mature osteoblasts [55]. BMP2 is also related to the promotion of osteoblastic differentiation of MSCs [56,57]. It was reported that *Bmp2* knockout affected prenatal and postnatal bone formation in mice [57] and expression of BMP2 was indispensable for fracture repair in mice [58]. Our results revealed that HPDLSCs-Exo promoted osteoblastic differentiation of osteoblast-like cells through upregulation of *ALP*, *BMP2*, *OCN*, and *OPN*, which are involved in different stages of osteoblastic differentiation, indicating that HPDLSCs-Exo have the ability to enhance osteoblastic differentiation at all stages.

Exosomes transmit signals that support bone remodeling by regulating cell differentiation, cell recruitment, and extracellular matrix mineralization in normal bones [59]. Previously, many kinds of cell sources were used for the isolation of MSCs-Exo, such as bone marrow [60], adipose [61], synovia [62], and umbilical cord [63]. Regardless of the cell source, all types of MSCs-Exo exhibited a promotion of bone healing [48], consistent with our results of HPDLSCs-Exo. When we compared the ability of HPDLSCs-Exo and bone-marrow-derived MSCs-Exo (BMSCs-Exo) to induce osteoblastic differentiation in osteoblast-like cells, Alizarin red S staining showed more mineralized cells in the HPDLSCs-Exo group than in the BMSCs-Exo group (Appendix A). A previous report demonstrated that HPDLSCs possessed a higher growth potential than BMSCs [64]. These results suggested that the HPDLSCs established in our previous study [35] may be a good cell source for exosomes with a high healing potential for bone defects.

Regarding the determination of exosomal concentration, previous in vitro reports with the mineralization assay demonstrated that the concentration of exosomes ranged from 1 to 200 µg/mL [65,66]. In our research, two concentrations of HPDLSCs-Exos—1 µg/mL and 2 µg/mL—were selected because concentrations less than 1 µg/mL did not show obvious effects on the differentiation of osteoblast-like cells in preliminary experiments. For in vivo reports about calvarial bone defects, the concentration of exosomes ranged from 10 to 1000 µg/mL [65,67]. Many studies reported applying over 100 µg/mL of exosomes for the healing of calvarial bone defects. In the present study, we observed that an exosomal concentration of 60 µg/mL exhibited good healing effects on calvarial bone defects whereas an exosomal concentration of 40 µg/mL did not show an obvious healing effect. These results suggested that HPDLSCs-Exo may be a useful tool for the healing of periodontal tissue including alveolar bone because they were effective at lower concentrations than seen in previous reports.

## 5. Conclusions

We successfully isolated exosomes from HPDLSCs, referred to as HPDLSCs-Exo, which exhibited the typical characteristics of exosomes. HPDLSCs-Exo promoted the migration and differentiation of osteoblast-like cells but not their proliferation. Furthermore, HPDLSCs-Exo enhanced the amount of newly formed bone at the site of calvarial bone defects in rats. These results suggest that HPDLSCs-Exo could be a new tool for the treatment of defective periodontal tissue, including alveolar bone, by cell-free therapy.

## Figures and Tables

**Figure 1 biomolecules-14-01455-f001:**
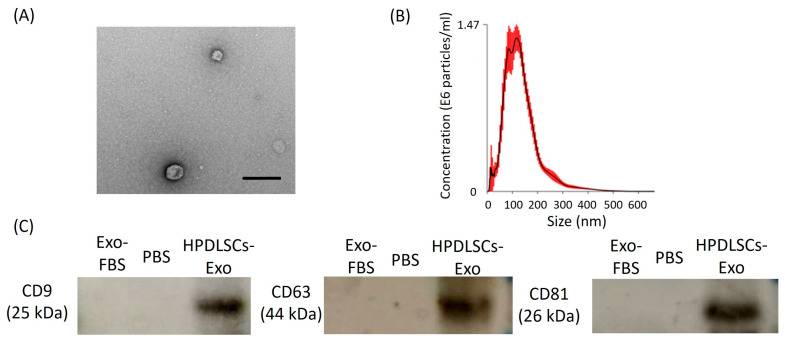
Isolation of human periodontal ligament stem-cell-derived exosomes (HPDLSCs-Exo). (**A**) An image of HPDLSCs-Exo was obtained by transmission electron microscopy. Scale bar = 200 nm. (**B**) Particle size distribution of HPDLSCs-Exo was analyzed by nano tracking analysis. Measurement of the particle size was performed five times and the average values were plotted. The mean value of the particles was 133 ± 6.4 nm and the mode (the most commonly occurring value) was 94 ± 21 nm. (**C**) Western blotting analysis showed the expression of CD9, CD63, and CD81 in HPDLSCs-Exo. Exosome-depleted FBS (Exo-FBS) and PBS were used as negative controls.

**Figure 2 biomolecules-14-01455-f002:**
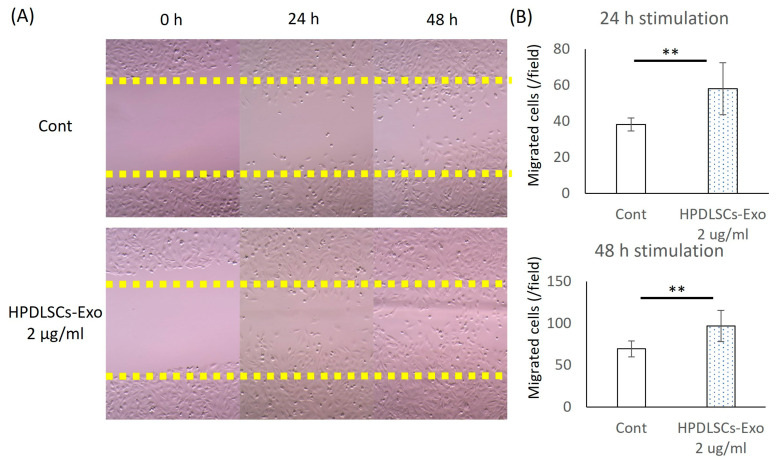
Effects of HPDLSCs-Exo on migration of Saos2 cells. (**A**) Saos2 cells were cultured in α-MEM containing 10% exosome-depleted-FBS (control) and control with 2 µg/mL of HPDLSCs-Exo. The scratch-wound healing was observed at three time points after the scratch (0 h, 24 h, and 48 h). *n* = 4. (**B**) Quantification of migrated cells was performed. The values are shown as the averages of migrated cell numbers per well. *n* = 4, ** *p* < 0.01.

**Figure 3 biomolecules-14-01455-f003:**
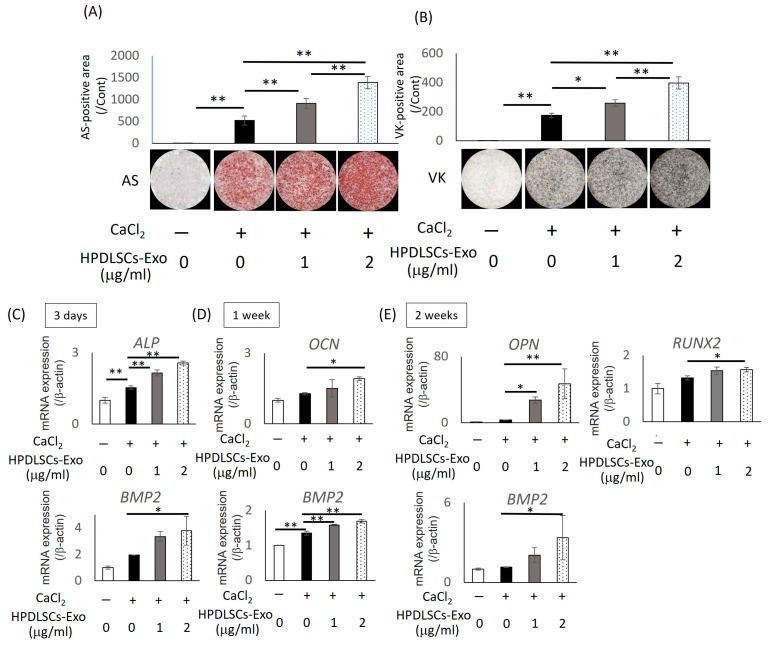
Effects of HPDLSCs-Exo on differentiation of Saos2 cells. Saos2 cells were cultured in α-MEM containing 10% exosome-depleted FBS (control), control containing 1 mM CaCl_2_ (Ca), and Ca with 1 or 2 µg/mL of HPDLSCs-Exo. (**A**,**B**) Alizarin red S staining (AS) and von Kossa staining (VK) were performed after 2 weeks of culture. AS-positive areas (**A**) and VK-positive areas (**B**) were quantified. * *p* < 0.05, ** *p* < 0.01, *n* = 3. (**C**–**E**) Saos2 cells were cultured in control, Ca, and Ca with 1 or 2 µg/mL of HPDLSCs-Exo for 3 days (**C**), 1 week (**D**), and 2 weeks (**E**). Quantitative RT-PCR was performed to analyze the expression of bone-related markers such as *alkaline phosphatase* (*ALP*), *bone morphogenetic protein 2* (*BMP2*), *osteocalcin* (*OCN*), *osteopontin* (*OPN*), and *runt-related transcription factor 2* (*RUNX2*). Normalization of gene expression was performed against β-actin expression, and the gene expression levels were shown as the fold increase relative to controls. * *p* < 0.05, ** *p* < 0.01, *n* = 3.

**Figure 4 biomolecules-14-01455-f004:**
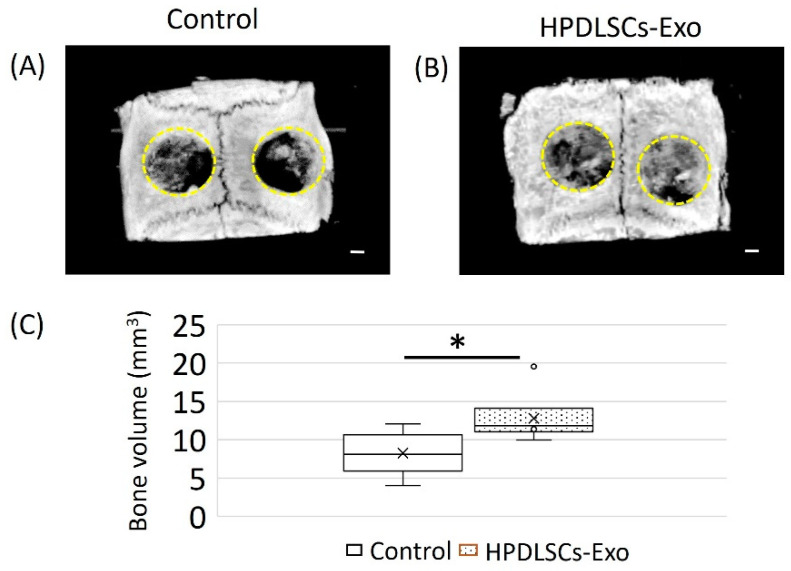
Effects of HPDLSCs-Exo on bone healing in rat calvarial bone defects. (**A**,**B**) Micro-CT images of defect sites in rat calvarial bone with or without HPDLSCs-Exo treatment. The rats (12 weeks old, male) were anesthetized and two critical-sized calvarial defects (5-millimeter diameter, yellow dotted circle) were created on either side of the calvarial bone using a dental trephine bur. Hydrogel with PBS (control) or hydrogel with 60 µg/mL of HPDLSCs-Exo (HPDLSCs-Exo) was injected into the defect sites. For each rat, both defect sides received control or HPDLSCs-Exo treatment. After 6 weeks of treatment, the samples were collected and images of control (**A**) and HPDLSCs-Exo (**B**) were obtained by micro-CT. Scale bars = 1 mm. (**C**) The areas of newly formed bone at the defect sites were quantified. *n* = 6, * *p* < 0.05.

**Figure 5 biomolecules-14-01455-f005:**
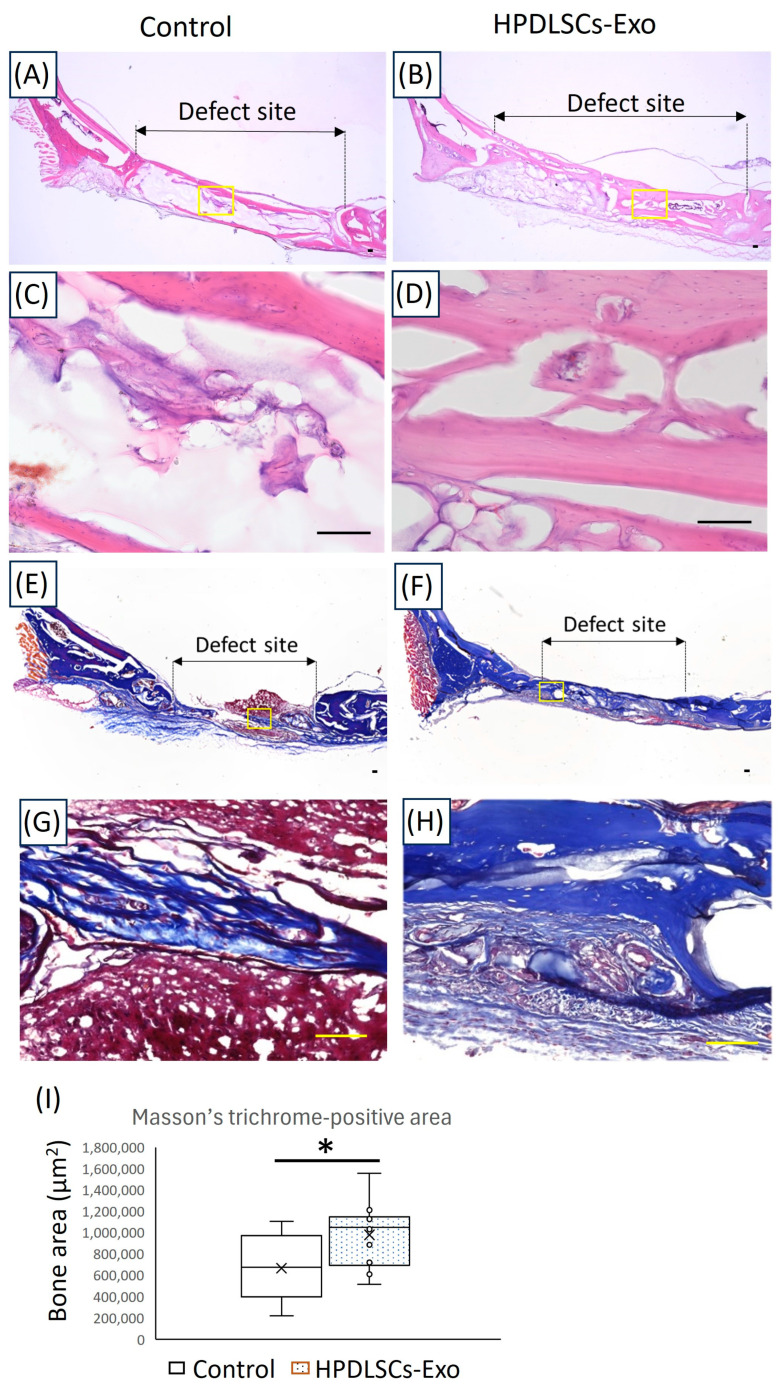
Histological analyses of the newly formed bone areas at the defect sites in rat calvarial bone. (**A**–**D**) Images of calvarial bone defect sites made in 12-week-old rats after 6 weeks of treatment with hydrogel made with PBS (control) or 60 µg/mL of HPDLSCs-Exo (HPDLSCs-Exo) were analyzed after HE staining. Panels (**C**,**D**) are magnified images of the yellow boxes in panels (**A**,**B**). Scale bars = 100 μm. (**E**–**H**) Images of calvarial bone defect sites from rats after 6 weeks of treatment with control or HPDLSCs-Exo were analyzed after Masson’s trichrome staining. Panels (**G**,**H**) are magnified images of the yellow boxes in panels (**E**,**F**). Scale bars = 100 μm. (**I**) The areas of newly formed bone at the defect sites in Masson’s trichrome staining (blue) were quantified. *n* = 5, * *p* < 0.05.

**Table 1 biomolecules-14-01455-t001:** Primer sequence, product size, and annealing temperature for quantitative RT-PCR.

Target Gene (Abbreviation)	Forward (Top) and Reverse (Bottom) Primer Sequences	Size of AMPLIFIED Products (bp)	Annealing Temperature (°C)	Sequence ID
b-actin	5′-ATTGCCGACAGGATGCAGA-3′	89	60	NM_001101.3
	5′-GAGTACTTGCGCTCAGGAGGA-3′			
ALP	5′-CTCTATCTTTGGTCTGGCCC-3′	154	60	NM_001177520.3
	5′-CTGCGCCTGGTAGTTGTTGT-3′			
BMP2	5′-TCCACTAATCATGCCATTGTTCA-3′	73	60	NM_001200.4
	5′-GGGACACAGCATGCCTTAGGA-3′			
OCN	5′-GTGCAGAGTCCAGCAAAGGT-3′	175	60	NM_199173.6
	5′-TCAGCCAACTCGTCACAGTC-3′			
OPN	5′-ACACATATGATGGCCGAGGTGA-3′	115	60	NM_000582.3
	5′-TGTGAGGTGATGTCCTCGTCTGT-3′			
RUNX2	5′-AACCCTTAATTTGCACTGGGTCA-3′	145	60	NM_001024630.3
	5′-CAAATTCCAGCAATGTTTGTGCTAC-3′			

## Data Availability

The original contributions presented in this study are included in the article/Appendix A; further inquiries can be directed to the corresponding author.

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
