# Peer review of "Exosomes from Human Periodontal Ligament Stem Cells Promote Differentiation of Osteoblast-like Cells and Bone Healing in Rat Calvarial Bone"

_biomolecules, 2024, doi:10.3390/biom14111455_

Round 1

Reviewer 1 Report

Comments and Suggestions for Authors

The study examines the effects of exosomes derived from human periodontal ligament stem cells (HPDLSCs-Exo) on osteogenic differentiation and bone healing. The research was conducted both in vitro, using human Saos-2 osteosarcoma cell cultures, and in vivo, utilizing a rat calvarial bone defect model. The authors successfully isolated and characterized exosomes from HPDLSCs, demonstrating their ability to promote mineralization in vitro, migration of osteosarcoma cells, and bone healing at defect sites in rats. These findings suggest that HPDLSCs-Exo could serve as a potential therapeutic tool for bone regeneration and periodontal tissue repair. While the results are promising and contribute valuable insights into the field of periodontal regeneration and bone repair, the study has several areas that require significant improvement before it can be considered for publication.

Please find the detailed, point-by-point comments below.

Major Comments:

1)       Saos-2 cells are derived from primary osteosarcoma and cannot be classified as pre-osteoblasts. It is, therefore, inappropriate to use the term "pre-osteoblasts" throughout the paper. Instead, the term "osteoblast-like cells" or another similar term should be used. For example, the statement from the discussion, “These results indicated that HPDLSCs-Exo is better able to induce osteoblastic differentiation of pre-osteoblasts than to promote their proliferation,” cannot be made based on data obtained using Saos-2 cells. The Saos-2 cell line is often considered the most representative model of osteoblasts, but it does not meet the criteria to be referred to as pre-osteoblasts. This issue has been well-discussed in several studies, such as Dvorakova et al., 2023 (https://www.ncbi.nlm.nih.gov/pmc/articles/PMC10069154/).

2)       It is unusual to induce osteogenic differentiation solely with CaCl2 without using dexamethasone and phosphate substrates in the medium. Could the authors provide references in the Materials and Methods section to other studies where CaCl2 alone was shown to induce Saos-2 mineralization without the use of classic osteogenic medium components?

3)       In Figures 1B and 1C, it is essential to include FBS processed in the same manner as the samples as a control. This will demonstrate that the FBS was not contaminated by particles (Fig. 1B), such as those that might originate from dust during analysis, and will ensure that no positive staining appears in western blotting in Fig. 1C. Additionally, it is unclear how many replicates of the western blotting shown in Figure 1C were performed. This information should be clearly stated.

4)       Raw data counts should be provided in the supplementary materials for Figure 2B. Currently, the figure compares control normalized to control and HPDLSCs-Exo normalized to the same control? It is unclear why this normalization is necessary in this context.

5)       The list of markers verified by qPCR to demonstrate osteogenic differentiation should include Runx2, a key regulator of osteogenic differentiation.

6)       In Figure 4C, there is an outlier in HPDLSCs-Exo group with better bone regeneration. Is this the same sample shown in Figure 4B? Are the differences still significant when the outlier with much better bone growth is excluded? Additionally, micro-CT images of all replicates should be shown in Figures 4A-B for complete transparency.

7)       The histological analysis is currently presented as representative images without any quantification. Automated and reproducible quantification analysis should be performed, as described in protocols such as those found in BioProtocols (https://bio-protocol.org/pdf/bio-protocol3629.pdf). For example, the extent of connective tissue/collagen (blue) in the tissue sections stained with Masson’s Trichrome could be quantified using image analysis tools like ImageJ, Photoshop, and Aperio.

Minor Points:

1)       Please clarify how and how frequently the absence of mycoplasma contamination was verified.

2)       The exact catalog numbers of antibodies used should be specified in the Materials and Methods section for clarity and reproducibility.

3)       In Materials and Methods section 2.6, the authors state that the number of migrated cells was counted manually but do not provide clear criteria for this process. The method of cell counting needs to be described in detail. It is also unclear why only three random fields per well were averaged rather than quantifying all fields.

4)       In Materials and Methods section 2.9, the authors mention that cDNA “was synthesized from 1 mg of the total RNA.” The use of 1 mg is unusual (though possible), as most protocols use 1 μg. Please confirm if this is not a typo.

5)       The statement, “Our results indicated that 2 µg/mL of HPDLSCs-Exo was enough to promote differentiation of pre-osteoblasts,” is not well supported by the data. The authors showed that HPDLSCs-Exo enhances the differentiation of Saos-2 cells under osteogenic differentiation conditions but does not independently “promote” it. Additionally, as previously mentioned, Saos-2 cells should not be referred to as “pre-osteoblasts.”

Comments on the Quality of English Language

The English is fine in general. Still, the minor grammatical adjustments and refinements in sentence structure would improve the overall quality before publication.

Author Response

Thank you very much for your suggestion.

Major Comments:

Q1. Saos-2 cells are derived from primary osteosarcoma and cannot be classified as pre-osteoblasts. It is, therefore, inappropriate to use the term "pre-osteoblasts" throughout the paper. Instead, the term "osteoblast-like cells" or another similar term should be used. For example, the statement from the discussion, “These results indicated that HPDLSCs-Exo is better able to induce osteoblastic differentiation of pre-osteoblasts than to promote their proliferation,” cannot be made based on data obtained using Saos-2 cells. The Saos-2 cell line is often considered the most representative model of osteoblasts, but it does not meet the criteria to be referred to as pre-osteoblasts. This issue has been well-discussed in several studies, such as Dvorakova et al., 2023

(https://www.ncbi.nlm.nih.gov/pmc/articles/PMC10069154/).

  1. Thank you very much for your suggestion. We changed the term of Saos-2 cells from ‘pre-osteoblasts’ to ‘osteoblast-like cells’ in the section of ‘Title’, ‘Abstract’, ‘Materials and Methods’ and ‘Discussion’.

Q2. It is unusual to induce osteogenic differentiation solely with CaCl2 without using dexamethasone and phosphate substrates in the medium. Could the authors provide references in the Materials and Methods section to other studies where CaCl2 alone was shown to induce Saos-2 mineralization without the use of classic osteogenic medium components?

  1. Thank you very much for your suggestion. We added the following references which CaCl2 alone was shown to induce mineralization of the cells in the Materials and Methods section (p4, line 172 - 173 in Materials and Methods).

  1. Bernar, A., et al., Optimization of the Alizarin Red S Assay by Enhancing Mineralization of Osteoblasts. Int J Mol Sci, 2022. 24(1).
  2. Maeda, H., et al., Mineral trioxide aggregate induces bone morphogenetic protein-2 expression and calcification in human periodontal ligament cells. J Endod, 2010. 36(4): p. 647-52.

Q3. In Figures 1B and 1C, it is essential to include FBS processed in the same manner as the samples as a control. This will demonstrate that the FBS was not contaminated by particles (Fig. 1B), such as those that might originate from dust during analysis, and will ensure that no positive staining appears in western blotting in Fig. 1C. Additionally, it is unclear how many replicates of the western blotting shown in Figure 1C were performed. This information should be clearly stated.

  1. Thank you very much for your suggestion. We added the results of western blotting analysis using FBS (We used exosome depleted-FBS in this study). The samples from FBS exhibited no positive staining (Fig. 1C). We confirmed these results three times. We added this information to the manuscript (p4, line 145 in Materials and Methods, p7, line 274 - 295 in Results and Supplemental Fig. 6).

Q4. Raw data counts should be provided in the supplementary materials for Figure 2B. Currently, the figure compares control normalized to control and HPDLSCs-Exo normalized to the same control? It is unclear why this normalization is necessary in this context.

  1. Thank you very much for your suggestion. We added the raw data counts of migrated cells in Supplemental Fig. 2. Furthermore, we showed the migrated cell numbers per field in the graphs rather than normalization in Fig. 2 (p8, line 311 - 317 in Results and Supplemental Fig. 2).

Q5. The list of markers verified by qPCR to demonstrate osteogenic differentiation should include Runx2, a key regulator of osteogenic differentiation.

  1. Thank you very much for your suggestion. The expression of RUNX2 was upregulated in HPDLSCs-Exo group compared with Ca group after 2 weeks of culture. We added this information to ‘Materials and Methods’, ‘Results’ and Fig. 3 (p5, line 202 - 208 in Materials and Methods, p8, line 325 - 330 in Results and Fig. 3).

Q6. In Figure 4C, there is an outlier in HPDLSCs-Exo group with better bone regeneration. Is this the same sample shown in Figure 4B? Are the differences still significant when the outlier with much better bone growth is excluded? Additionally, micro-CT images of all replicates should be shown in Figures 4A-B for complete transparency.

  1. Thank you very much for your question. The outlier in HPDLSCs-Exo group is the same sample shown in Figure 4B. However, there were significant differences even when the outlier sample was excluded as shown in Supplemental Fig. 3B. Furthermore, we showed all micro-CT images of Control group and HPDLSCs-Exo group in Supplemental Fig. 3A (Supplemental Fig. 3).

Q7. The histological analysis is currently presented as representative images without any quantification. Automated and reproducible quantification analysis should be performed, as described in protocols such as those found in BioProtocols (https://bio-protocol.org/pdf/bio-protocol3629.pdf). For example, the extent of connective tissue/collagen (blue) in the tissue sections stained with Masson’s Trichrome could be quantified using image analysis tools like ImageJ, Photoshop, and Aperio.

  1. Thank you very much for your suggestion. We performed the quantification of connective tissue/collagen (blue color) areas in tissue sections stained with Masson’s Trichrome staining by a Biozero digital microscope (Keyence Corporation, Osaka, Japan). We added this information to the manuscript (Fig. 5I).

Minor Points:

Q1. Please clarify how and how frequently the absence of mycoplasma contamination was verified.

  1. Thank you very much for your question. We used Takara PCR Mycoplasma Detection Set (Takara Bio Inc., Shiga, Japan). In addition, we performed this experiment using the samples after 3 days, 1 week and 2 weeks of culture. We added this data in Supplemental Fig.5.

Q2. The exact catalog numbers of antibodies used should be specified in the Materials and Methods section for clarity and reproducibility.

  1. Thank you very much for your suggestion. We added the exact catalog numbers of antibodies (p4, line 137 - 140 in Materials and Methods).

Q3. In Materials and Methods section 2.6, the authors state that the number of migrated cells was counted manually but do not provide clear criteria for this process. The method of cell counting needs to be described in detail. It is also unclear why only three random fields per well were averaged rather than quantifying all fields.

  1. Thank you very much for your question. We performed the migration assay based on the previous studies and these reports selected three random fields per well with three samples (1, 2). Furthermore, we increased the field and sample numbers (four fields per well with four samples), and added the references of these reports and the detail of quantification. (p4, line 148 - 159 in Materials and Methods)

  1. Adachi, O., et al., Decorin Promotes Osteoblastic Differentiation of Human Periodontal Ligament Stem Cells. Molecules, 2022. 27(23).
  2. Sugii, H., et al., Effects of Activin A on the phenotypic properties of human periodontal ligament cells. Bone, 2014. 66: p. 62-71.

Q4. In Materials and Methods section 2.9, the authors mention that cDNA “was synthesized from 1 mg of the total RNA.” The use of 1 mg is unusual (though possible), as most protocols use 1 μg. Please confirm if this is not a typo.

  1. We appreciate your comment very much. We wrote it incorrectly. We changed the concentration from ‘1 mg’ to ‘1 µg’ (p5, line 191 in Materials and Methods).

Q5. The statement, “Our results indicated that 2 µg/mL of HPDLSCs-Exo was enough to promote differentiation of pre-osteoblasts,” is not well supported by the data. The authors showed that HPDLSCs-Exo enhances the differentiation of Saos-2 cells under osteogenic differentiation conditions but does not independently “promote” it. Additionally, as previously mentioned, Saos-2 cells should not be referred to as “pre-osteoblasts.”

  1. Thank you very much for your suggestion. We omitted the following sentences from our manuscript in ‘Discussion’, “Our results indicated that 2 µg/mL of HPDLSCs-Exo was enough to promote differentiation of pre-osteoblasts”.

Reviewer 2 Report

Comments and Suggestions for Authors

·         MSCs of different tissue sources are currently being considered as an attractive tool for cell therapy. The current MS is focused on the characterization of exosomes derived from highly promising populations of MSCs  - periodontal ligament stem cells (PDLSCs) originated from  dental stem cells.  This study is complex, and the authors have demonstrated that PDLSCs-derived exosomes have osteogenic activity both in vitro and in vivo.

·                  The primary issue with this study can be addressed as follows: In the Introduction, the authors have effectively outlined the challenges posed by "irreversible destruction of the periodontal tissue,  which is a complex structure supporting teeth and is composed of periodontal ligament (PDL), cementum, and alveolar bone”. Moreover, they have sought to identify an additional therapeutic option for these cases using PDLSCs-Exo. In vitro testing has shown that PDLSCs-Exo can stimulate osteodifferentiation in pre-osteoblasts. Please clarify why calvarial bone was used for the in vivo assay if the goal was to help periodontal tissue. It would be more reasonable to use an alveolar bone defect model.  

·                This is a descriptive study based on standard protocols applied in modern cell biology experiments.

·                 Did the authors try other than 2ug/ml Exo concentration for migration assay?

·               The Discussion lacks a coherent structure and is somewhat loose.  Please note that the data from the Results section has been repeated several times.

Author Response

Thank you very much for your suggestion.

Q1. The primary issue with this study can be addressed as follows: In the Introduction, the authors have effectively outlined the challenges posed by "irreversible destruction of the periodontal tissue, which is a complex structure supporting teeth and is composed of periodontal ligament (PDL), cementum, and alveolar bone”. Moreover, they have sought to identify an additional therapeutic option for these cases using PDLSCs-Exo. In vitro testing has shown that PDLSCs-Exo can stimulate osteodifferentiation in pre-osteoblasts. Please clarify why calvarial bone was used for the in vivo assay if the goal was to help periodontal tissue. It would be more reasonable to use an alveolar bone defect model.  

  1. Thank you very much for your question. As you pointed out, the final goal of our research is regeneration of defected periodontal tissue which is composed of PDL, cementum alveolar bone. The tooth loss is primarily due to the resorption of supporting bone. Regeneration of resorped bone is essential for regeneration of defected periodontal tissue. Thus, we focused on the regeneration of defected bone in this study. We added this information to ‘Introduction’. Furthermore, we will try to perform the animal experiments using rat periodontal tissue defected model in future study, because we have already established this animal model in the previous report (1). (p1, line 33 - 35 in Introduction)

  1. Kaneko, H., et al., Inhibition of c-Jun N-terminal kinase signaling promotes osteoblastic differentiation of periodontal ligament stem cells and induces regeneration of periodontal tissues. Arch Oral Biol, 2022. 134: p. 105323.

Q2.  Did the authors try other than 2ug/ml Exo concentration for migration assay?

  1. Thank you very much for your question. We performed migration assay using the 5 ug/ml HPDLSCs-Exo. This concentration also promoted migration of Saos-2 cells compared with Cont group, but there was no change between 2 ug/ml HPDLSCs-Exo group and 5 ug/ml HPDLSCs-Exo group. We added this data in Supplemental Fig. 2.

Q3. The Discussion lacks a coherent structure and is somewhat loose.  Please note that the data from the Results section has been repeated several times.

  1. Thank you very much for your suggestion. We improved ‘Discussion’ section and removed the repeated data from the Results section in Discussion section. (p11 - 13, line 446 – 450, 488 -512 in Discussion)

Round 2

Reviewer 1 Report

Comments and Suggestions for Authors

The authors addressed most of my comments and improved the manuscript significantly. Still, two points need further improvement:

1) Western blots are of poor quality in general, but the CD9 is especially unclear. For CD9, it is unclear whether the signal is caused by an artifact or specific staining. I strongly suggest repeating CD9 WB to obtain better quality.

2) Q3 is partially resolved, but additional experiments are necessary. As I mentioned before, the absence of particles within an expected size in the FBS needs to be shown in Fig 1B as a negative control. As the authors mentioned, they used exosome-depleted FBS, but particles within this size might also come from dust and plastic. Therefore, only FBS negative control is necessary for nano-tracking analysis.

Author Response

Thank you very much for your suggestion.

Comments:

Q1. Western blots are of poor quality in general, but the CD9 is especially unclear. For CD9, it is unclear whether the signal is caused by an artifact or specific staining. I strongly suggest repeating CD9 WB to obtain better quality.

A. Thank you very much for your suggestion. We changed all images of western blotting (CD9, CD63, and CD81) in Fig. 1 and Supplemental Fig. 6.

Q2. Q3 is partially resolved, but additional experiments are necessary. As I mentioned before, the absence of particles within an expected size in the FBS needs to be shown in Fig 1B as a negative control. As the authors mentioned, they used exosome-depleted FBS, but particles within this size might also come from dust and plastic. Therefore, only FBS negative control is necessary for nano-tracking analysis."

A. Thank you very much for your suggestion. In our study, the nano-tracking analysis was outsourced to an outside contractor (FUJIFILM Wako Pure Chemical Industries Ltd.). Thus, we could not perform this analysis until deadline date. In addition, we purchased exosome-depleted FBS from System Biosciences. That company performed nano-tracking analysis using the Exo-FBS. The results showed the absence of particles within an expected size in the Exo-FBS. We added the picture of nano-tracking analysis and website.

From Exo-FBS | System Biosciences

Reviewer 2 Report

Comments and Suggestions for Authors

The responses provided by the authors are deemed to be satisfactory.  

Author Response

Thank you very much for your comment.

Round 3

Reviewer 1 Report

Comments and Suggestions for Authors

As soon as nano-tracking analysis was outsourced, I agree that it would be to demanding to reproduce the whole nano-tracking analysis. Therefore the paper might be accepted in the present form.

Of note: I would like to emphasize again the importance of incorporating a FBS control in nano-tracking analysis that verifies the absence of contamination by dust particles during sample preparation, rather than relying solely on the fact that such particles were absent in the initial FBS. I strongly suggest that the authors include this type of control in future studies to ensure the accuracy and integrity of the results.